# Increased Chymase-Positive Mast Cells in High-Grade Mucoepidermoid Carcinoma of the Parotid Gland

**DOI:** 10.3390/ijms24098267

**Published:** 2023-05-05

**Authors:** Hiromi Nishimura, Denan Jin, Ichita Kinoshita, Masataka Taniuchi, Masaaki Higashino, Tetsuya Terada, Shinji Takai, Ryo Kawata

**Affiliations:** 1Department of Otorhinolaryngology—Head and Neck Surgery, Osaka Medical and Pharmaceutical University, Takatsuki-City 569-8686, Osaka, Japan; hiromi.nishimura@ompu.ac.jp (H.N.); ichita.kinoshita@ompu.ac.jp (I.K.); masataka.taniuchi@ompu.ac.jp (M.T.); masaaki.higashino@ompu.ac.jp (M.H.); tetsuya.terada@ompu.ac.jp (T.T.); ryo.kawata@ompu.ac.jp (R.K.); 2Department of Innovative Medicine, Graduate School of Medicine, Osaka Medical and Pharmaceutical University, Takatsuki-City 569-8686, Osaka, Japan; shinji.takai@ompu.ac.jp

**Keywords:** salivary glands, mucoepidermoid carcinoma, chymase, mast cell, angiotensin II

## Abstract

It has long been known that high-grade mucoepidermoid carcinoma (MEC) has a poor prognosis, but the detailed molecular and biological mechanisms underlying this are not fully understood. In the present study, the pattern of chymase-positive mast cells, as well as chymase gene expression, in high-grade MEC was compared to that of low-grade and intermediate-grade MEC by using 44 resected tumor samples of MEC of the parotid gland. Chymase expression, as well as chymase-positive mast cells, was found to be markedly increased in high-grade MEC. Significant increases in PCNA-positive cells and VEGF gene expression, as well as lymphangiogenesis, were also confirmed in high-grade MEC. Chymase substrates, such as the latent transforming growth factor-beta (TGF-β) 1 and pro-matrix metalloproteinase (MMP)-9, were also detected immunohistologically in high-grade MEC. These findings suggested that the increased chymase activity may increase proliferative activity, as well as metastasis in the malignant condition, and the inhibition of chymase may be a strategy to improve the poor prognosis of high-grade MEC of the parotid gland.

## 1. Introduction

Major salivary glands consist of parotid, submandibular, and sublingual glands. Salivary glands are made up of different kinds of cells and cancers can start in any of these cell types. Therefore, many different types of cancer can develop in each type of salivary gland. Parotid carcinomas represent 70% of all salivary gland malignancies and mucoepidermoid carcinoma (MEC) has been reported to be the most common histological type [1]. Based on histological scoring, the degree of necrosis, mitoses, atypical nuclei, and size of the cystic component in the tumor tissues, MEC is typically classified into three histological grades, i.e., low, intermediate, and high [2,3]. Generally, histological grading is an important prognostic factor, and the mortality rate usually tends to increase in higher-grade carcinoma and decrease in lower-grade carcinoma [4,5]. In fact, McHugh et al. reported that the five-year disease-specific survival (DSS) rate of parotid MEC was significantly lower in the high-grade MEC population (67.0%, *p* < 0.001) [4]. On the other hand, they found no significant difference in five-year DSS rates between the low-grade and intermediate-grade MEC populations (98.8% vs. 97.4%). So far, the detailed molecular and biological mechanisms underlying the differential survival rates that exist between these groups are not fully understood.

Recently, research on the relationship between chymase and malignant tumor pathology has attracted a great deal of attention. For example, Kinoshita et al. recently reported that the number of chymase-positive mast cells, as well as chymase gene expression, was increased markedly in carcinoma ex pleomorphic adenoma (CXPA) of the parotid gland when compared to the benign parotid tumor, pleomorphic adenoma (PA) [6]. Chymase-positive cells were also significantly increased in human lung and gastric cancers [7,8], indicating the importance of chymase in cancer. Chymase was first demonstrated as an angiotensin (Ang) II-forming enzyme in cardiovascular tissue ([9,10] and, since then, the various roles of chymase have gradually emerged. For example, chymase was reported to enzymatically activate the latent transforming growth factor-beta (TGF-β) 1 and pro-matrix metalloproteinase (MMP)-9 to their active forms [11,12]. As is known, TGFβ1 activation through the suppression of T cells’ anticancer function [13] decreases the immune surveillance function against cancer generation. On the other hand, to initiate cancer metastasis cancer cells must become motile and invasive, as well as intravasate. All of these steps require the breakdown of cell–cell and cell–extracellular matrix (ECM) contacts, as well as the basement membrane of the vasculature or lymphatic vessels; MMP-9 is a powerful degrading enzyme in these processes [14]. Interestingly, it was also reported that the chymase-mediated Ang II can also promote both angiogenesis and lymphangiogenesis [15] through the chymase-Ang II- vascular endothelial growth factor (VEGF)-dependent pathway. Taken together, one can see that the mast cell-derived chymase may promote not only cancer development but also cancer metastasis through the activation of TGFβ1, MMP-9, and VEGF. Therefore, the present study aimed to evaluate if there were significant differences in the number of chymase-positive mast cells and chymase expression among high-grade, intermediate-grade, and low-grade conditions by examining paraffin-embedded samples of salivary gland MECs. Since the survival rates of the intermediate-grade and low-grade MEC populations are equally high [4], the resected samples from the intermediate-grade and low-grade MECs were grouped together as a good prognosis group (I–L group), and they were compared with the high-grade MECs as the poor prognosis group (H group) in the present study.

## 2. Results 

### 2.1. Subject Profile

Table 1 shows the demographic and clinicopathological features of the enrolled patients. 

As indicated in Table 1, the present study enrolled a total of 44 patients, with 18 high-grade MEC patients (H group). As mentioned above, since the clinical prognosis of the intermediate-grade and low-grade MECs was similar, the remaining 26 samples were combined as one group (I–L group) in the present study. The male-to-female ratio was double in the high-grade MEC group, whereas the proportion of females in the low-grade MEC group was more than double that of males, and the male-to-female ratio was similar in the intermediate-grade MEC group. The age of the high-grade MEC group ranged from 42 to 85 years for males and from 42 to 82 years for females, whereas the low-grade MEC group tended to be young, with ages ranging from 19 to 59 years for males and from 19 to 69 years for females. Regardless of grade, the most common anatomic location of MEC was the superficial lobe of the parotid glands. In the present study, rates of facial nerve paralysis, lymph node metastasis, and recurrence were significantly higher in the H group than in the I–L group, but there was little difference in pain or tenderness between the two groups (Table 1).

### 2.2. The Histological Features of MEC

Figure 1 shows the representative HE-stained and Azan Mallory-stained cross-sections from the I–L and H groups. Black hashes indicate cystic lesions and black asterisks indicate solid tumor sites. As can be seen, the cystic lesions were frequently observed in the I–L group, whereas the H group was occupied mainly by solid tumor tissues. The majority of solid tumor tissues in the I–L group were separated into tumor islands by the collagen-deposited (blue color) stroma, and the contours of tumor islands were mostly clear and smooth. On the other hand, the solid tumor tissues in the H group were separated irregularly by intermittent, thin, collagen-deposited stroma, and the contours of tumor islands were neither clear nor even. 

### 2.3. The Identification of Fibroblasts and Cancer-Associated Fibroblasts

Figure 2 shows representative vimentin and α-SMA-immunostained serial cross-sections from the I–L and H groups. The yellow frames of vimentin and α-SMA immunostaining correspond to the positions of the yellow frames of Azan-Mallory staining, positioned in the boundary of the solid tumor. Vimentin is a marker protein for mesenchymal origin cells that is mainly expressed in fibroblasts and myofibroblasts, as well as endothelial cells [16]. Though α-SMA as a contractile protein is mainly expressed in contractile vascular smooth muscle cells, it is also expressed in myofibroblasts after the phenotypic change from fibroblasts has occurred [17]. Therefore, the proportions of fibroblasts and myofibroblasts among tumors can be calculated with the combination of vimentin and α-SMA immunostaining. As shown in Figure 2, the vimentin-positive cells in the I–L group were mainly distributed at the stroma site. As shown in the yellow frame of Azan Mallory staining, the blue color at the boundary of the solid tumor was dark, suggesting the deposition of a large amount of collagen in that area. The yellow frame photograph of vimentin staining on the right shows the area inside the yellow frame of Azan Mallory staining. In addition, α-SMA immunostaining performed in the serial section adjacent to the vimentin immunostaining confirmed that the α-SMA-positive cells were partially overlapped with the vimentin-positive cells, indicating the mixed presence of fibroblasts and myofibroblasts in the boundary of solid tumors. A similar pattern was also observed in the H group. As shown in the bottom of Figure 2, almost all vimentin-positive cells overlapped with the α-SMA-positive cells, indicating that myofibroblasts are the major cellular portion in these sites. In the present study, the expression ratio of myofibroblasts tended to increase in the H group, which may indicate the presence of more cancer-associated fibroblasts (CAFs) in the malignant condition. The concept of CAFs is new, and the prevalence of this population was reportedly correlated with a poor prognosis for malignancy [18].

### 2.4. Identification of Type of Mast Cells

Figure 3 shows representative toluidine blue staining, as well as the calculated number of mast cells in the I–L and H groups. As shown in the bar graph of Figure 3, the numbers of mast cells in the H group tended to increase compared to the I–L and H groups (*p* = 0.057).

Figure 4 shows the chymase immunostaining and the calculated numbers of chymase-positive cells, as well as the gene expression of chymase in the tumor tissues from the I–L and H groups. As shown in the bar graph, the number of chymase-positive cells was significantly higher in the H group than in the I–L group. The gene expression level of chymase also tended to be higher in the H group than in the I–L group (*p* = 0.054). Similarly, the number of tryptase-positive cells, as well as tryptase gene expression, were significantly higher in the H group than in the I–L group (Figure 5).

According to the contents of neutral proteases in secretory granules, two types of human mast cells have been recognized. If mast cells contain both tryptase and chymase in their secretory granules, these cells belong to the MC_TC_ subtype, and if they contain only tryptase, these cells belong to the MC_T_ subtype [19]. In the present study, staining with toluidine blue, chymase, and tryptase was performed on three serial cross-sections to clarify which types of mast cells were expressed in MECs of the parotid glands. The diameter of mast cells is usually about 20 μm. Therefore, if the cross-sections are made to be within 4-μm-thick, the rate of appearance of the same mast cell in three respective serial cross-sections is comparatively high. 

As can be seen in Figure 6, the locus of mast cells, confirmed by toluidine blue staining, was largely overlapped with that of chymase-positive and tryptase-positive cells (HPF 1000×, yellow arrows), indicating that mast cells were the main cellular source for these neutral proteases in the MEC. Moreover, the mean number of mast cells, chymase-positive, and tryptase-positive cells were found to be similar in the respective two groups, suggesting that the MC_TC_ type is the major expressing mast cell in both the I–L and H groups.

### 2.5. Examination of Proliferative Cells, Neovascularization, and Lymphangiogenesis

Figure 7 shows representative PCNA immunostaining, as well as the estimated number of PCNA-positive cells in the I–L and H groups. PCNA is highly expressed in proliferating cells, especially during the G1 and S phases of the cell cycle [20], and an increase in PCNA-positive cells in tumor tissues may indicate a poor prognosis [21]. As can be seen in the bar graph of Figure 7, the PCNA-positive cells were significantly higher in the H group than in the I–L group.

Figure 8 shows representative podoplanin and vWF immunostaining, as well as the gene expression levels of VEGF in the I–L and H groups. Podoplanin is specifically expressed in lymphatic cells [22,23], and vWF is a marker of vascular endothelial cells [24]; these are useful markers for differentiating lymphatic and blood vessels in tumor tissues. VEGF is a product of macrophages and tumor cells and not only promotes blood vascular angiogenesis, but also lymphangiogenesis [25,26]. In comparison with the I–L group, the gene expression level of VEGF tended to be higher in the H group. Figure 8, displaying podoplanin and vWF staining, shows the serial cross-sections from the I–L and H groups. As can be seen in the upper imaging photograph, the areas of podoplanin-positive circle staining (blue arrow) are few in the I–L group. However, the areas of vWF-positive circle staining (green arrow) in the lower imaging photograph are observed in large numbers. In Figure 8, areas of podoplanin and vWF staining in the right panel are also the images from the serial cross-sections of the H group. In contrast to the I–L group, the areas of podoplanin-positive circle staining (blue arrow) in the H group are observed in large numbers, whereas the areas of vWF-positive circle staining (green arrow) in the H group are few. 

### 2.6. Identification of MMP-9, MMP-2, TGFβ-1 and SCF-Positive Cells

Figure 9 shows the representative staining results for HE and Azan Mallory in the H group. The images of immunostaining for MMP-9, MMP-2, TGFβ1, and SCF are also serial cross-sections from the same individual. As can be seen in these photographs, MMP-9-positive staining could be observed in both the tumor-like cells (enlarged photograph of black frame) and fibroblast-like spindle cells (enlarged photograph of green frame). On the other hand, MMP-2-positive staining was only found in the tumor-like cells (enlarged photograph of black frame) and their staining densities were lighter than those of the areas of MMP-9-positive staining. Areas of SCF immunostaining were also found in the tumor-like cells (enlarged photograph of black frame), as well as in the fibroblast-like spindle cells (enlarged photograph of green frame), whereas TGFβ1 immunostaining was mainly found in the tumor-like cells (enlarged photograph of black frame).

## 3. Discussion

Salivary gland tumors account for about 5% of all neoplasms of the head and neck, with most of them occurring in the parotid glands, which are the largest of the three sets of major salivary glands [27]. Of parotid gland tumors, malignant tumors account for 20%, and MEC is the most common histological type [1]. MEC is usually classified into three histological grades [2,3] and mortality rates among the three grades differ greatly. According to the results of the treatment at our department, the five-year DSS for the intermediate-grade and low-grade patients both reached above 95%, whereas the five-year DSS for high-grade patients decreased to 53.8% [28]; these data are similar to previous reports [4]. The poor prognosis of MEC patients can be estimated based on the combination of the histological grade and the clinical stage. Of these criteria, the presence or absence of distant metastases affects the survival rate the most [29]. Distant metastases may occur through either vascular or lymphatic vessels and the mobility of tumor cells in the metastatic processes is the most critical condition. In general, whether normal or tumor cells, these cells adhere to the extracellular matrix (ECM) and cannot migrate. Therefore, detachment from the restricted ECM to become a mobile tumor cell is essential for distant metastasis. Then, the tumor cells need to intravasate to the vasculature or lymphatic vessels and become survivable circulating tumor cells (CTCs) until colonization at a distant organ through extravasation. In this step, breaking the basement membrane of the vasculature or lymphatic vessels is also indispensable. In this way, many metastatic processes need to break down ECM proteins, and MMPs are believed to be the main degradative enzymes. As mentioned in the introduction, the mast cell-derived chymase is not only a powerful Ang II-forming enzyme but is also an activator for both latent TGF-β1 and pro-MMP-9 [11]. Interestingly, the activated TGF-β1 was also able to induce pro-MMP-9 protein expression in metastatic tumor cells, indicating the presence of an additive effect on MMP-9 action after chymase activation [30]. On the other hand, it has been reported that chymase, through the activation of protease-activated receptor (PAR)-2, increases the gene expression of MMP-2 [31]. Although MMP-2 and MMP-9 share the ability to degrade denatured collagen, they can also degrade laminin and elastin [32]. Previously, we also found that chymase itself could degrade fibronectin [33]. Considering the above direct and indirect actions of chymase, the increased chymase activity may not only detach cell–cell and cell–extracellular matrix (ECM) adhesions but may also break down the integrity of the basement membrane in the vasculature or lymphatic vessels under certain conditions. In the present study, the chymase-positive mast cells, as well as chymase gene expression, were markedly increased in the H group (Figure 4). Areas of MMP-9 and MMP-2 positive staining were also detectable in the tumor cells and fibroblast-like cells (Figure 9). As indicated in Table 1, the prevalence of lymph node metastases was significantly higher in the H group than in the I–L group. These findings may suggest the presence of aggressive chymase-dependent MMP activation in the high-grade MEC. In the present study, SCF immunostaining was confirmed in tumor-like cells and fibroblast-like spindle cells. SCF not only acts as a major chemotactic factor for mast cells and their progenitors, but also elicits cell–cell and cell–substratum adhesion, facilitates proliferation, and sustains the survival, differentiation, and maturation of mast cells [34]. Interestingly, it has been reported that mast cell-derived chymase [35] could enzymatically cleave the membrane-bound SCF to release the bioactive form of SCF, suggesting that the increase in chymase-positive cells in the H group may be a result of chymase and tumor cell surface SCF interactions.

In the present study, there was also a marked increase in PCNA-positive tumor cells, as well as in the gene expression levels of VEGF in the H group (Figure 7 and Figure 8). Moreover, consistent with the increased VEGF gene expression levels, a large amount of lymphangiogenesis was observed in the H group (Figure 8). Although no photograph of angiogenesis in the H group has been provided, aggressive angiogenesis was also found in this group. VEGF can promote both angiogenesis and lymphangiogenesis, and may increase the frequency of blood and lymphatic metastases. Angiogenesis can also increase nutrient supply and contribute to tumor growth. Therefore, a marked increase in PCNA-positive tumor cells may be associated with the presence of more aggressive angiogenesis in the H group. 

On the other hand, TGF-β1-positive staining was also detectable in the H group. TGF-β1 is not only a key player in fibrosis in some organs, but it also acts as the tumor microenvironment (TME) and as a carcinomatous transformation regulator in cancer pathophysiology [36]. The tumor mass consists not only of a heterogeneous population of cancer cells, but also the so-called TME, which includes infiltrating host cells, secreted factors, and extracellular matrix proteins, as well as various tumor-associated cells, such as CAFs. These microenvironments seem to be important in cancer invasion and metastasis, as well as in angiogenesis, and TGF-β1 is an important mediator for building up the TME. For example, TGF-β1 plays an important role in cancer migration due to its mediation of CAF contractility and MMP secretion, where CAFs produce MMPs that destroy the structure of the TME architecture [37,38]. In the present study, α-SMA-positive cells occupied the major cellular portion in the H group (Figure 2), suggesting that CAFs are abundant in such malignant environments. On the other hand, it has been reported that increased TGF-β1 levels not only suppress T cells’ anticancer function [13], but they also inhibit T-cell proliferation by decreasing the expression of interleukin-2 [39]; thus, the immune surveillance by T cells tends to decrease. Therefore, a significant increase in PCNA-positive tumor cells in the H group may also result from the decrease in immune surveillance by the activated TGF-β1. 

Henceforth, the true roles of chymase in the pathology of parotid gland carcinoma should be confirmed in animal models or human clinical studies by using a chymase-specific inhibitor. Unfortunately, since an animal model that reflects the pathology of parotid gland cancer has not yet been established, we could not verify this issue in the present study. However, previous animal studies seem to suggest the important roles of the renin-angiotensin system on cancer growth and metastasis. For example, specific AT1R blockade reportedly suppressed VEGF production, resulting in reduced tumor angiogenesis and slow progression of tumor growth in a mouse prostate cancer (C4-2 cells) xenograft model [40]. Interestingly, it was reported that the angiotensin activation ability in CT26 mouse colon cancer cells was dependent mainly on the renin-chymase pathway, rather than the renin-ACE pathway [41]. Moreover, when compared with untreated mice, treatment with chymostatin, a chymase inhibitor, not only largely suppressed liver metastasis of CT26 cells, but also significantly improved the survival rate of CT26 cells in a spleen-injected model. Taken together, these findings strongly suggest that an increase in chymase activity and chymase-positive mast cells in the TME might worsen the prognosis of cancer.

### Limitations

The present study only focused on the features of the histological distribution and the degree of chymase expression between the two groups. However, since mast cells contain several mediators, including tryptase and carboxypeptidase A3, these are also reported to be involved in tumor pathophysiology. Therefore, the distribution characteristics of proteases in the MEC should also be clarified in the future.

## 4. Materials and Methods

### 4.1. Sample Collection and Grouping

Of the surgically resected parotid gland tumors in our hospital (1999–2020), 44 formalin-fixed and paraffin-embedded blocks of MEC were obtained from the Department of Pathology. Per the criteria of the American Forces Institute of Pathology (AFIP) [2,3], these samples included 18 high-grade, 9 intermediate-grade, and 17 low-grade MEC samples. As mentioned above, the good prognosis group contained 26 MEC samples from both the intermediate-grade and low-grade populations, which constituted the I–L group in the present study. On the other hand, the poor prognosis group contained the 18 high-grade MEC samples and constituted the H group in the present study. This study was performed following the ethical principles regarding human experimentation in the Declaration of Helsinki and was approved by the Research Ethics Committee of Osaka Medical and Pharmaceutical University (authorization number: 2866-1).

### 4.2. General Histological and Immunohistological Studies

For histological and immunohistological staining, 4-μm-thick, serial cross-sections were prepared from paraffin blocks of low-grade, intermediate-grade, and high-grade MEC samples using a microtome (LITORATOMU, REM-710, Yamato Koki Kogyo Ltd., Saitama, Japan). The first serial cross-sections from each of the paraffin blocks were stained with hematoxylin and eosin (HE) to observe their general structures. For the second sections, Azan Mallory staining was performed to identify fibrotic areas. HE staining and Azan Mallory staining were performed per the standard staining protocols. Toluidine blue staining was performed on the fourth sections to identify mast cell distribution. In brief, deparaffinized sections were immersed in 0.5% toluidine blue solution (pH 4.8) for around 15 min, fractionated with 0.5% glacial acetic acid solution, and mounted after drying. 

The third and fifth sections were used to show the distribution of chymase and tryptase using an anti-chymase antibody (mouse monoclonal antibody against human mast cell chymase, 2D11G10D, 1:1000 dilution; a gift from Suzuki, Katakura Industries Co., Saitama, Japan) and anti-tryptase antibody (M7052, 1:800 dilution; Dako, Glostrup, Denmark), respectively. The sixth sections were used to stain for TGF-β1 (ARP37894-P505, 1:100 dilution; Aviva Systems Biology, San Diego, CA, USA). The seventh and eighth sections were used to stain for MMP-9 (RB1590-P1, 1:50 dilution; Lab Vision Co., CA, USA) and MMP-2 (RB1537-P1, 1:50 dilution; Lab Vision Co.). To evaluate mesenchymal cellular components (such as fibroblasts and myofibroblasts) within the tumor tissues, vimentin (1:70 dilution; Cell Signaling Technology, Danvers, MA, USA) and α-smooth muscle actin (α-SMA) (1:200 dilution; Dako) immunostainings were performed on the ninth and tenth sections. The eleventh sections were used to stain for von Willebrand factor (vWF) (1:100 dilution; Dako) to evaluate the degree of angiogenesis. Podoplanin is a lymphatic marker because the expression of podoplanin has been detected in lymphatic but not blood vascular endothelium, and it is useful as the marker of tumor-associated lymphangiogenesis [22]. To evaluate lymphangiogenesis in the MEC, podoplanin immunostaining (11629-1-AP, 1:100 dilution; Proteintech Group, Rosemont, IL, USA) was performed on the twelfth sections in the present study. The thirteenth sections were used to stain for SCF (26582-1-AP, 1:200 dilution; Proteintech Group). To identify the growth activity of the tumors, proliferating cell nuclear antigen (PCNA) (1:100 dilution; Dako) immunostaining was also performed on the fourteenth sections. 

Immunostaining with the abovementioned antibodies was performed following the protocols described elsewhere [6,42]. In brief, deparaffinized sections were incubated with the respective antibodies overnight at 4 °C, followed by a reaction with components from a labeled streptavidin-biotin peroxidase kit (Dako LSAB kit; Dako, Carpinteria, CA, USA). Thereafter, these sections were incubated with 3-amino-9-ethylcarbazole (AEC) for color development, counterstained with hematoxylin, and, finally, mounted with cover glasses. The three densest areas in each cross-section were counted in a high-power field (HPF:200×) to count the cellular number, and the average value of the three areas was used for statistical analysis.

### 4.3. Real-Time Reverse Transcriptase Polymerase Chain Reaction (RT-PCR)

A value of ten sheets of the 10-µm-thick paraffin sections were cut out from the respective formalin-fixed, paraffin-embedded tissue blocks using the microtome. Then, the RNA was extracted from these tissues, using methods described elsewhere [6]. Briefly, total RNA was extracted by the protocol provided in the total RNA isolation kit (ISOGEN PB Kit, Nippon Gene Co., Ltd., Tokyo, Japan). Total RNA (2.5 µg) was transcribed into cDNA with SuperScript VILO (Invitrogen, Carlsbad, CA, USA). Then, mRNA levels were measured by RT-PCR on a Stratagene Mx3000P (Agilent Technologies, San Francisco, CA, USA) using Taq-Man fluorogenic probes. RT-PCR primers and probes for tryptase, chymase, vascular endothelial growth factor (VEGF), and glyceraldehyde-3-phosphate dehydrogenase (GAPDH) were designed by Roche Diagnostics (Tokyo, Japan). 

The primers were as follows: 5′-gatgctgagcctgctgct-3′ (forward) and 5′-gacgatacccgcttgctg-3′ (reverse) for tryptase, 5′-cattaacgggttcagttccag-3′ (forward) and 5′-agcaggaagggtcggttc-3′ (reverse) for chymase, 5′-gcagcttgagttaaacgaacg-3′ (forward) and 5′-ggttcccgaaaccctgag-3′ (reverse) for VEGF, and 5′-agccacatcgctcagacac-3′ (forward) and 5′-gcccaatacgaccaaatcc-3′ (reverse) for GAPDH. The probes were as follows: 5′-ctgcccca-3′ for tryptase, 5′-cagaggaa-3′ for chymase, 5′-ctccttcc-3′ for VEGF, and 5′-tggggaagg-3′ for GAPDH. The mRNA levels of tryptase, chymase, and VEGF were normalized to those of GAPDH. 

### 4.4. Statistical Analysis

All numerical data are expressed as means ± SEM. Significant differences in mean values between the two groups were evaluated with Student’s *t*-test. Significant differences in the presence or absence of the symptom of pain, facial paralysis, lymph node metastasis, and recurrence between the two groups were evaluated with the Chi-squared test. In all analyses, a *p*-value less than 0.05 was considered significant.

## 5. Conclusions

In comparison with the intermediate-grade and low-grade MECs, the numbers of chymase-positive mast cells and PCNA-positive tumor cells, as well as chymase and VEGF gene expressions, were all markedly increased in the high-grade MEC. Given chymase’s powerful enzymatic effects on Ang I, latent TGF-β1, and pro-MMPs, increased chymase activity may increase proliferation, as well as metastasis, in the malignant condition. Therefore, the inhibition of chymase may be a strategy to improve the poor prognosis of high-grade MEC.

## Figures and Tables

**Figure 1 ijms-24-08267-f001:**
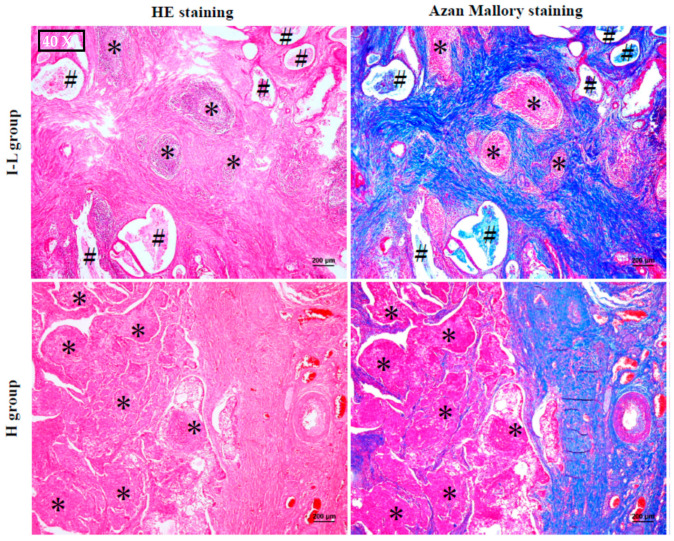
Representative HE and Azan Mallory staining in the cross-sections from the I–L and H groups. Black hashes indicate cystic lesions and black asterisks indicate solid tumor tissues.

**Figure 2 ijms-24-08267-f002:**
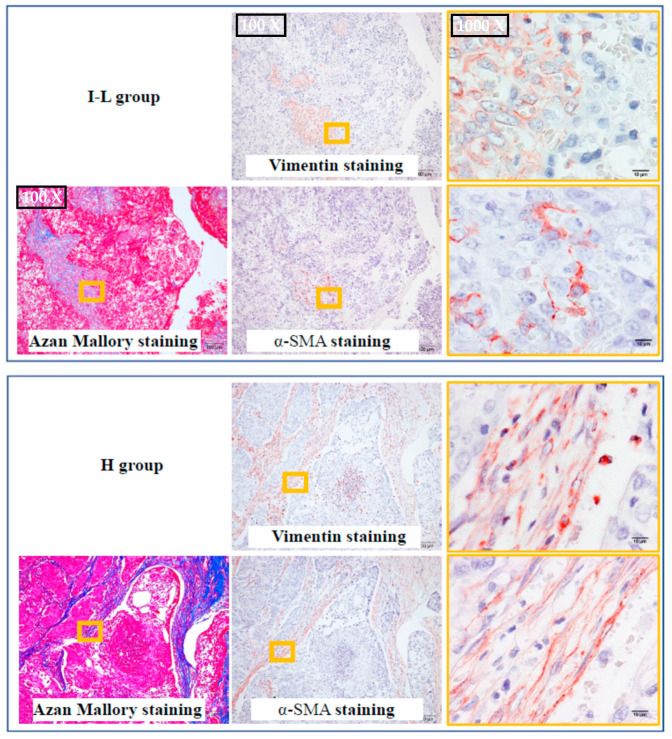
Representative vimentin and α-SMA immunostaining in the serial cross-sections from the I–L and H groups. The yellow frames of the vimentin and α-SMA immunostaining correspond to the positions of the yellow frames of Azan-Mallory staining.

**Figure 3 ijms-24-08267-f003:**
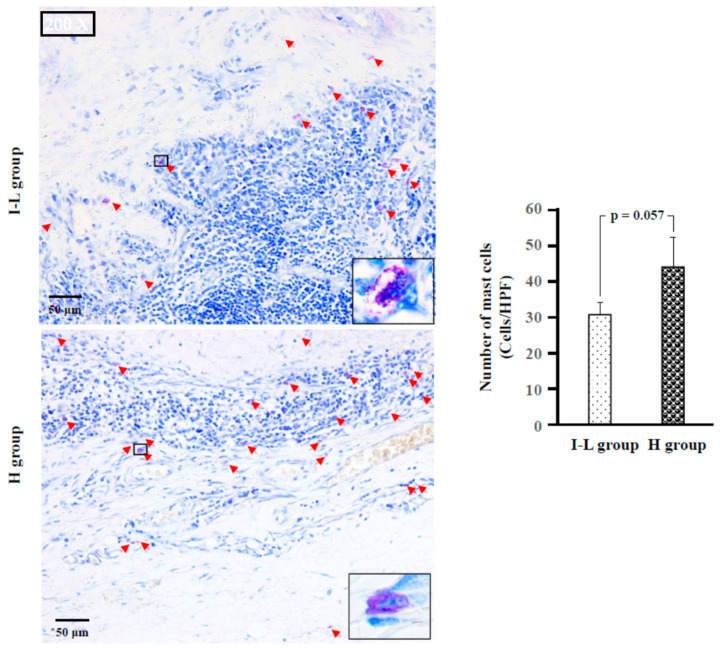
Representative toluidine blue staining, as well as the calculated number of mast cells in the I–L and H groups. Red arrows indicate mast cells. As can be seen in these photographs, the cytoplasm of mast cells is stained purple with toluidine blue.

**Figure 4 ijms-24-08267-f004:**
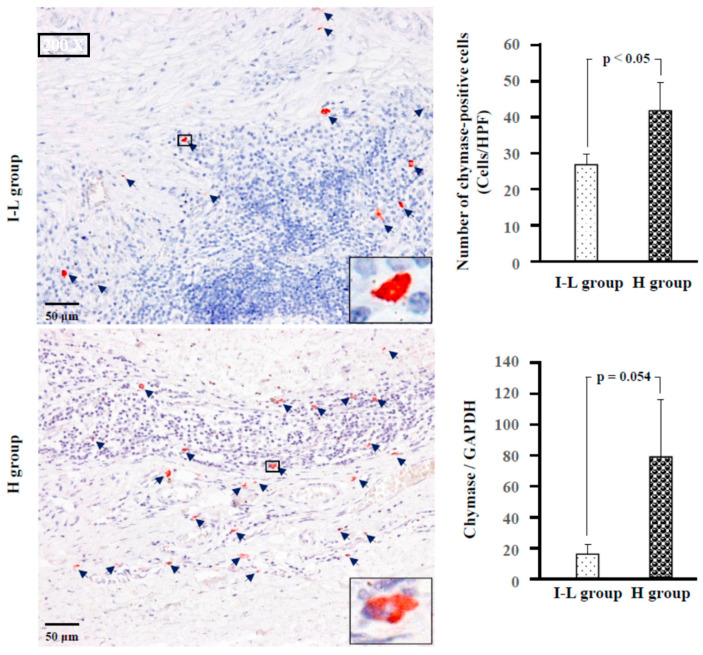
Representative chymase immunostaining and the calculated number of chymase-positive cells, as well as chymase gene expression in the I–L and H groups. Navy blue arrows indicate chymase-positive cells.

**Figure 5 ijms-24-08267-f005:**
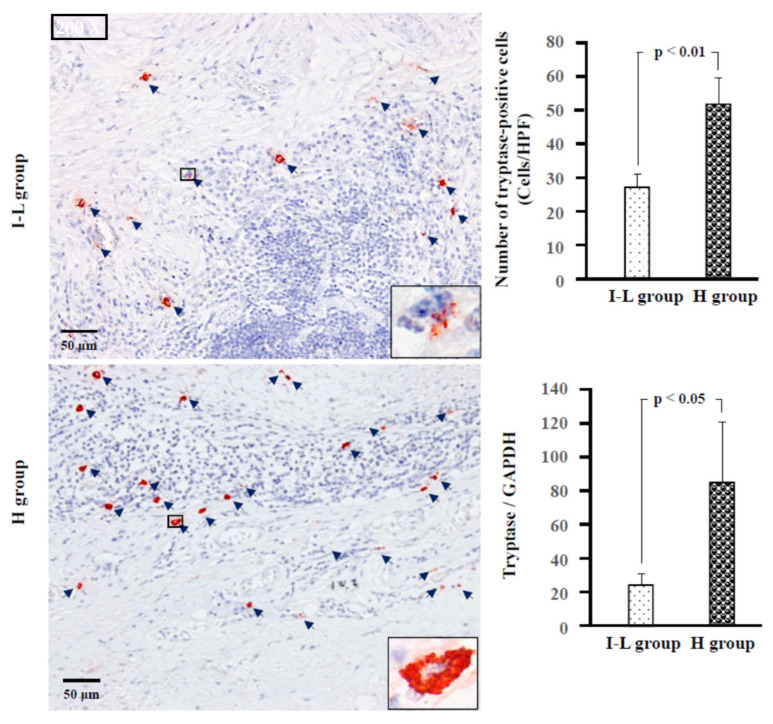
Representative tryptase immunostaining and the calculated number of tryptase-positive cells, as well as tryptase gene expression in the I–L and H groups. Navy blue arrows indicate tryptase-positive cells.

**Figure 6 ijms-24-08267-f006:**
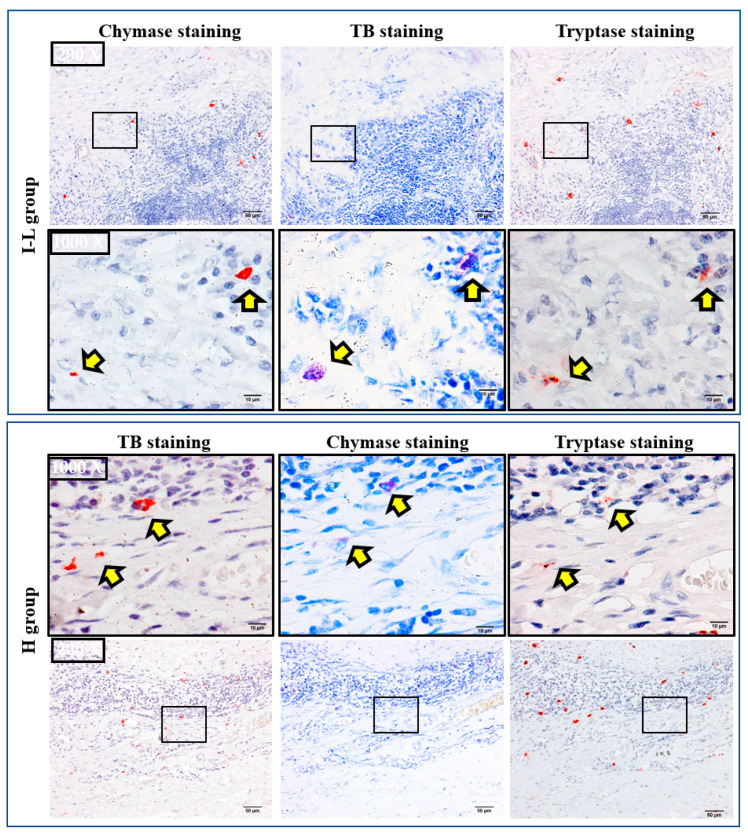
Representative toluidine blue, chymase, and tryptase staining in the serial cross-sections positioned before and after toluidine blue staining. Yellow arrows indicate mast cells, chymase-positive, and tryptase-positive cells, respectively. Mast cells confirmed by toluidine blue staining are almost in the same position as the chymase-positive and tryptase-positive cells (HPF 1000×), indicating that mast cells are the main cellular source in these tumors.

**Figure 7 ijms-24-08267-f007:**
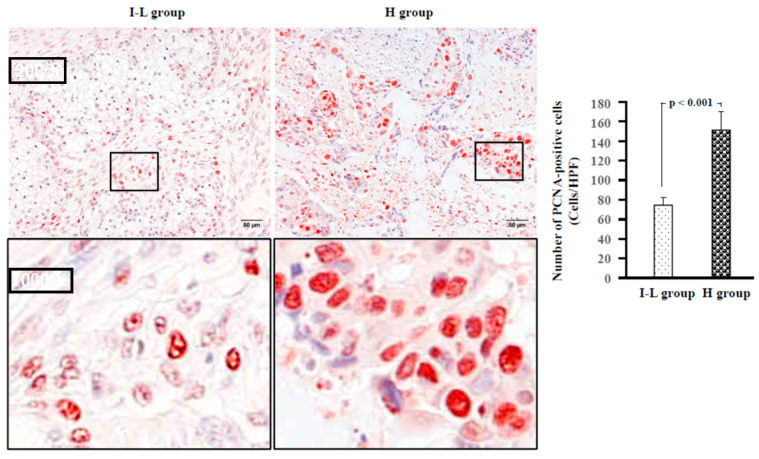
Representative PCNA immunostaining, as well as the estimated numbers of PCNA-positive cells in the I–L and H groups.

**Figure 8 ijms-24-08267-f008:**
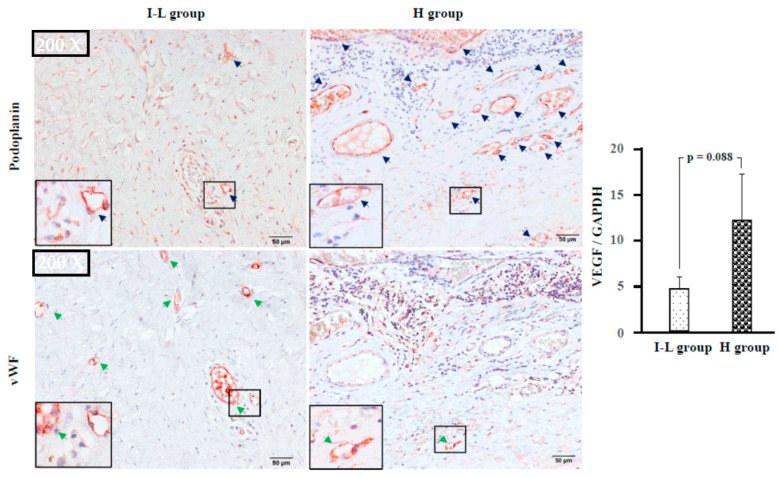
Representative podoplanin and vWF immunostaining, as well as gene expression levels of VEGF in the I–L and H groups. Navy blue and green arrows indicate lymphatics and blood vessels, respectively.

**Figure 9 ijms-24-08267-f009:**
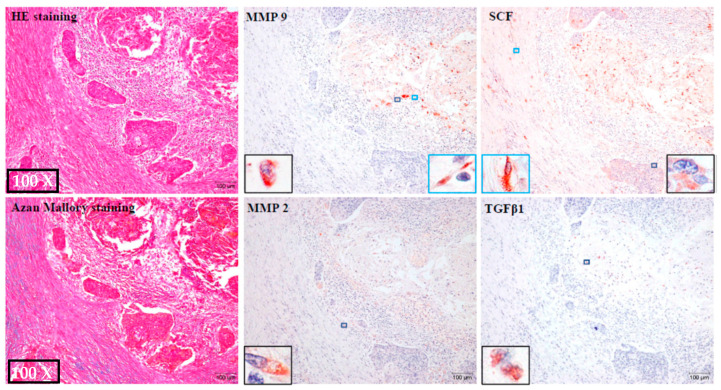
Representative HE and Azan Mallory staining from the H group. The images of MMP-9, MMP-2, TGFβ1, and SCF immunostaining are also serial cross-sections from the same individual.

**Table 1 ijms-24-08267-t001:** Demographic and clinicopathological features of the enrolled patients.

		High-Grade (n = 18)	Intermediate-Grade (n = 9)	Low-Grade (n = 17)	*p* Value
Sex	Male (Age range)	12 (42–85)	4 (22–72)	5 (19–59)	*p* < 0.05
	Female (Age range)	6 (42–82)	5 (23–79)	12 (19–69)	
Anatomic location of parotid tumor	Superficial	14	8	10	*p* = 0.83
	Others	4	1	7	
Symptom of pain	Yes	11	3	8	*p* = 0.22
	No	7	6	9	
Facial paralysis	Yes	3	0	0	*p* < 0.05
	No	15	9	17	
Lymph node metastasis	Yes	11	2	1	*p* < 0.001
	No	7	7	16	
Recurrence	Yes	8	0	2	*p* < 0.01
	No	10	9	15	

*p* value listed above is the result of statistical analysis that was calculated between the H and I-L groups. H group contain 18 high-grade MEC. I-L group contain 26 MEC samples from both the intermediate-grade and low-grade populations.

## Data Availability

The data that support the findings of this study are available on request from the corresponding author, Denan Jin (E-mail: denan.jin@ompu.ac.jp).

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
