# Peer review of "Increased Chymase-Positive Mast Cells in High-Grade Mucoepidermoid Carcinoma of the Parotid Gland"

_ijms, 2023, doi:10.3390/ijms24098267_

Round 1

Reviewer 1 Report

 In the work of Hiromi Nishimura et al. "Increased chymase-positive mast cells in high-grade mucoepidermoid carcinoma of the parotid gland", a study was conducted to analyze the involvement of one of the specific mast cell proteases, chymase, in the pathogenesis of a tumor disease. To solve this problem, the authors use interesting morphological approaches, including staining and analysis of serial sections to be able to interpret chymase colocalization in the same loci of tissue sections with other targets, in particular, tryptase, polyanions (metachromasia), PCNA, TGF- β1, MMP-9, MMP-2, α-SMA, etc.

In general, the work deserves support. However, the reviewer has a number of key questions, the answers to which are necessary to obtain more reliable and understandable information about the results of the study. Moreover, the conclusions of the work on the effects of chymase in its current form require more convincing evidence or additional discussion.

1. The authors are trying to analyze the ratio of different cells/targets in the tumor tissue. For example, tryptase and chymase. The authors write (p. 113): "Therefore, the proportions of fibroblasts and myofibroblasts among tumors can be calculated with the combination of vimentin and α‐SMA immunostaining." Why did the authors not use the technology of double immunolabeling to solve such problems, simultaneously detecting two or more targets on the same section? This would greatly simplify the analysis and make it possible to obtain more objective results.

2. It remains completely unclear from the article which tumor-associated MCs the authors studied: intratumoral or peritumoral? It is necessary to give an explanation.

3. Page 157. The author needs to indicate how the mast cell count was carried out, which algorithm of the planimetric approach was used to obtain quantitative data.

4. Line 127. Figure 2. The author needs to check the expression of SMA in the figure, and more precisely indicate where the myofibroblasts are located.

5. Line 170. Figure 6. The quality of micrographs needs to be significantly (!) Improved. It is currently impossible to prove that the authors present a similar tissue locus on serial sections - very poor image quality. In addition, the scale is completely invisible in the micrographs.

6. Line 184. How was PCNA calculated?

7. The authors do not fully reveal the role of chymase in oncogenesis. You can learn more about this in the reviews (1. Hellman L, Akula S, Fu Z, Wernersson S. Mast Cell and Basophil Granule Proteases - In Vivo Targets and Function. Front Immunol. 2022 Jul 5;13:918305. doi: 10.3389/ fimmu 2022.918305 PMID: 35865537;PMCID: PMC9294451 2. Atiakshin D, Buchwalow I, Tiemann M. Mast cell chymase: morphofunctional characteristics Histochem Cell Biol 2019 Oct;152(4):253-269 doi: 10.1007/ (S00418-019-01803-6. Epub 2019 Aug 8. PMID: 31392409.)

8. Why do the authors associate the results of their work only with mast cell chymase? The authors need to consider the potential role of carboxypeptidase A3 (CPA3) in the discussion, since the development of a number of biological effects of chymase depends on it (see works: 1. Hellman L, Akula S, Fu Z, Wernersson S. Mast Cell and Basophil Granule Proteases - In Vivo Targets and Function Front Immunol 2022 Jul 5;13:918305 doi: 10.3389/fimmu.2022.918305 PMID: 35865537; PMCID: PMC9294451 2. Pejler G, Knight SD, Henningsson F, Wernersson S. Novel insights into the biological function of mast cell carboxypeptidase A Trends Immunol 2009 Aug;30(8):401-8 doi: 10.1016/j.it.2009.04.008 Epub 2009 Jul 28 PMID: 19643669 3 Atiakshin D, Kostin A , Trotsenko I, Samoilova V, Buchwalow I, Tiemann M. Carboxypeptidase A3-A Key Component of the Protease Phenotype of Mast Cells Cells 2022 Feb 6;11(3):570 doi: 10.3390/cells11030570 PMID: 35159379; PMCID: PMC8834431.).

9. What is the reason for the conclusion that an increase in the activity of chymase and chymaso-positive mast cells in the tumor microenvironment can worsen cancer prognosis? According to the authors, the content of tryptase-positive MCs also increased, and tryptase, in turn, is known to have very active properties in relation to oncogenesis, including epigenetic effects (see articles: 1. Hellman L, Akula S, Fu Z, Wernersson S. Mast Cell and Basophil Granule Proteases - In Vivo Targets and Function. Front Immunol. 2022. 2. Atiakshin D, Buchwalow I, Samoilova V, Tiemann M. Tryptase as a polyfunctional component of mast cells. Histochem Cell Biol. 2018 May ;149(5):461-477 doi: 10.1007/s00418-018-1659-8 Epub 2018 Mar 12 PMID: 29532158 3. Melo, F.R.; Vita, F.; Berent-Maoz, B.; Levi Schaffer, F.; Zabucchi, G.; Pejler, G. Proteolytic histone modification by mast cell tryptase, a serglycin proteoglycan-dependent secretory granule protease J Biol Chem 2014, 289, 7682-7690, doi:10.1074/jbc.M113 546895. 4. Rabelo Melo, F.; Santosh Martin, S.; Sommerhoff, C. P.; Pejler, G. Exosome-mediated uptake of mast cell tryptase into the nucleus of melanoma cells: a novel axis for regulating tumor cell proliferation and gene expression. Cell Death Dis 2019, 10, 659, doi:10.1038/s41419-019-1879-4. 5. Alanazi, S.; Rabelo Melo, F.; Pejler, G. Tryptase Regulates the Epigenetic Modification of Core Histones in Mast Cell Leukemia Cells. Front Immunol 2021, 12, 804408, doi:10.3389/fimmu.2021.804408.

The authors need to consider this issue in the discussion.

Author Response

Dear Reviewer 1                       20230502

Thank you very much for your valuable comments. We have revised the discussion based on your suggestions, as follows:

Response to comment 1

As you rightly mentioned, double immunostaining is the most useful method for obtaining more objective results, such as proving that chymase and tryptase, as well as vimentin and α-SMA, originate in the same cells. In fact, in order to perform this double staining, we repeated the staining process several times using different fluorescent-labeled antibodies against chymase and tryptase, but with only limited success. As explained in the present paper, since the diameter of mast cells is usually about 20 μm, creation of 4-μm-thick cross-sections would result in a comparatively high rate of appearance of the same mast cell in three successive serial cross-sections. Therefore, we believe that the present results are accurate to some extent, although not as accurate as double staining.

Responses to comments 2 and 3

Response to comment 4

In accordance with your suggestion, a description of the myofibroblast expression site was added to the results, as follows (lines 125-128):

As shown in the yellow frame of Azan Mallory staining, the blue color at the boundary of the solid tumor was dark, suggesting the deposition of a large amount of collagen in that area. The yellow frame photo of vimentin staining on the right shows the area inside the yellow frame of Azan Mallory staining.

Response to comment 5

In accordance with your recommendation, we posted higher quality pictures in this revised manuscript. Both the low magnification and high magnification (1000x) images in Fig. 6 are serial sections. Although the higher magnification (1000x) looks slightly off, these are undoubtedly serial sections.

Response to comment 6

Like the calculation of mast cell number, the three most-dense areas in a high-power field of each cross-section were counted (HPF: 200x) to determine the PCNA number, and the average value of the three areas was used for statistical analysis.

Response to comment 7

Thank you very much for providing useful information on chymase cancer formation. Although we have not elaborated on all our views on chymase oncogenesis, some are presented in this paper. We speculate that chymase might be involved in carcinogenesis and cancer metastasis mainly by reducing immunosurveillance via activation of TGF-beta and by promoting MMP- and VEGF-mediated angiogenesis. We have already argued these points in this paper.

Response to comments 8 and 9

As you suggested, mast cells contain several mediators, including tryptase and carboxypeptidase A3, and the relationship between malignant tumors and these proteases have also been largely reported. Of course, we do not intend to deny the possibility of their involvement in MEC at all. However, since too much focus on one paper can be confusing, this point was added to the discussion as a shortcoming of this study as follows (lines 342-347):

Thank you again for your helpful comments. We believe that our paper has substantially improved with this revision.

Thank you very much for your kind consideration.

Best regards,

Denan Jin, M.D., Ph.D.

Department of Innovative Medicine, Graduate School of Medicine, Osaka Medical and Pharmaceutical University, 2-7 Daigaku-machi, Takatsuki, Osaka 569-8686, Japan.

                     TEL: +81-72-683-1221 (Ext2141)

                     FAX: +81-72-684-6730

                     E-mail: denan.jin@ompu.ac.jp

Reviewer 2 Report

This is an interesting paper, certainly deserving publication.

The relationship between chymase positive mast cells and the increased expression of VEGF, TGF-beta, MMPS and other stromal constitutents seems to be directly proportional to the number of mast cells in the tumor tissue. Anyhow, just to remain in the topic of salivary gland tumors, it should be commented that also adenolymphomas ( Warthin's tumors), most of which are entirely benign, are annedotically mast-cell-rich tumors, so it would be stimulating to compare the number of infiltrating mast-cells in these two lesions, just to underline that more often than thought, cancer is not a unifactorial disease.

The paper needs minor professional workup of the text.

Author Response

Dear Reviewer 2                     20230502

Thank you very much for your valuable comments. The present study only compared the expression of chymase and the number of chymase-positive mast cells among different types of mucoepidermoid carcinomas involving the parotid glands. As you pointed out, this study would have benefitted from a comparison of the expression characteristics of chymase in benign tumors versus mucoepidermoid carcinomas, which is important information to support our conclusion. However, at the time of starting this study, we did not know which benign tumor should be selected as the target group. As you pointed out, since Warthin's tumors might be a good target group for mucoepidermoid carcinomas, we plan to conduct a comparative study of this tumor and mucoepidermoid carcinomas in the near future.

Our manuscript was edited by a language editing company before its submission.

Thank you again for your valuable suggestions.

Best regards,

Denan Jin, M.D., Ph.D.

Department of Innovative Medicine, Graduate School of Medicine, Osaka Medical and Pharmaceutical University, 2-7 Daigaku-machi, Takatsuki, Osaka 569-8686, Japan.

                     TEL: +81-72-683-1221 (Ext2141)

                     FAX: +81-72-684-6730

                     E-mail: denan.jin@ompu.ac.jp
